# Mortality in Different Mountain Sports Activities Primarily Practiced in the Summer Season—A Narrative Review

**DOI:** 10.3390/ijerph16203920

**Published:** 2019-10-15

**Authors:** Hannes Gatterer, Martin Niedermeier, Elena Pocecco, Anika Frühauf, Martin Faulhaber, Verena Menz, Johannes Burtscher, Markus Posch, Gerhard Ruedl, Martin Burtscher

**Affiliations:** 1Institute of Mountain Emergency Medicine, EURAC Research, 39100 Bolzano, Italy; hannes.gatterer@eurac.edu; 2Department of Sport Science, University of Innsbruck, 6020 Innsbruck, Austria; martin.niedermeier@uibk.ac.at (M.N.); Elena.Pocecco@uibk.ac.at (E.P.); Anika.Fruehauf@uibk.ac.at (A.F.); Verena.Menz@uibk.ac.at (V.M.); Markus.Posch@uibk.ac.at (M.P.); Gerhard.Ruedl@uibk.ac.at (G.R.); 3Austrian Society for Alpine and High-Altitude Medicine, 6020 Innsbruck, Austria; Martin.Faulhaber@uibk.ac.at; 4Laboratory of Molecular and Chemical Biology of Neurodegeneration, École Polytechnique Fédérale de Lausanne, 1015 Lausanne, Switzerland; johannes.burtscher@epfl.ch

**Keywords:** mountain sports, risk, mortality, death risk, hiking, biking, paragliding, trekking, climbing, high altitude

## Abstract

Millions of people engage in mountain sports activities worldwide. Although leisure-time physical activity is associated with significant health benefits, mountain sports activities also bear an inherent risk for injury and death. However, death risk may vary across various types of mountain sports activities. Epidemiological data represent an important basis for the development of preventive measures. Therefore, the aim of this review is to compare mortality rates and potential risk factors across different (summer) mountain sports activities. A comprehensive literature search was performed on the death risk (mortality) in mountain sports, primarily practiced during the summer season, i.e., mountain hiking, mountain biking, paragliding, trekking, rock, ice and high-altitude climbing. It was found that the death risk varies considerably between different summer mountain sports. Mortality during hiking, trekking and biking in the mountains was lower compared to that during paragliding, or during rock, ice or high-altitude climbing. Traumatic deaths were more common in activities primarily performed by young adults, whereas the number of deaths resulting from cardiovascular diseases was higher in activities preferred by the elderly such as hiking and trekking. Preventive efforts must consider the diversity of mountain sports activities including differences in risk factors and practitioners and may more particularly focus on high-risk activities and high-risk individuals.

## 1. Introduction

The popularity of mountain sports activities is strongly increasing all over the world and particularly in the Alps. Millions of people are practicing one or more recreational sports in mountainous areas during the summer and the winter seasons [1,2]. 

Although it is indisputable that leisure-time physical activity is generally associated with health benefits [3], especially outdoor activities, they also bear an inherent risk of injury and even death [1]. Death is the most severe consequence of a misadventure and death rates may be considered as essential indicators for the hazardous nature of a certain type of sports activity. 

On the one hand, knowledge on mortality rates may enable comparing the fatality risk between sports and thus may help one to decide which specific sport activities to select according to the own risk tolerance. It was previously shown that the motivation to engage in sports with a certain element of risk (e.g., adventure sports) is multifaceted [4] and knowledge plays an important role in risk management strategies [5]. Thus, knowledge on mortality rates might result in abstaining from engagement in high-risk activities. 

On the other hand, such data represent an important stimulus to identify risk factors and mechanisms of fatal events. This knowledge may constitute the basis for the development of effective preventive measures [6,7,8]. Since mortality and risk factors might change over the years, e.g., due to a change in number of persons at risk [9], due to changes in equipment [10], due to updated knowledge [11], or due to preventive efforts (e.g., initiated by mountaineering associations [12]), the information on the year, and when the data was collected, should be considered. Describing secular trends may help to evaluate the potential effects of such changes/measures.

Therefore, this review aims to compare mortality rates and, where available, potential risk factors between mountain sports activities primarily practiced during the summer season.

## 2. Materials and Methods 

The literature search was performed in the following databases using a cut-off date of June 2019: Pubmed/Medline, Web of Science, Science direct, Scopus, and Sport Discus. The following keywords were used: death risk, mortality, mountain hiking, mountain biking, paragliding, trekking, rock climbing, ice climbing, mountaineering, and high-altitude climbing. Titles, abstracts, and relevant full-text articles have been assessed by the authors using the following specific inclusion criteria: original articles or review articles describing the death risk (mortality) and, where available, potential risk factors for mortality in mountain sport activities that are primarily practiced during the summer season as specified by the keywords. To the best of our knowledge, a categorization of all mountain sports is absent in scientific literature. Therefore, we included relevant mountain sports following accident statistics [13] and suggestions by the Austrian Alpine Association [14]. Reference lists of articles were also reviewed to ensure relevant studies were included. Additionally, data from national databases and reliable online sources were used for analysis. 

All relevant articles specifically for the mountain sports activities were reviewed by two authors responsible for each mountain sports activity: mountain hiking (MF and MB), mountain biking (EP and MB), paragliding (AF and HG), trekking (HG and MB), mountain, rock and ice climbing (MN and MB) and high-altitude climbing (MB and HG), respectively. Data extraction was conducted by the first author mentioned for each mountain sports activity. Primarily, fatality numbers related to the population at risk, and/or to the number of injuries/accidents, and/or specific information on exposure time were extracted from the selected studies, national reports and webpages. Descriptive statistics were reported in order to get some measure of comparison.

To provide an overview on mortality trends from the Austrian Alps, we additionally used the unique database of the “Österreichisches Kuratorium für Alpine Sicherheit”. These data included all fatalities that occurred in the Austrian mountains from 1997 to 2017 [13,15]. For more detailed information see [16]. As a rough surrogate for the numbers of persons at risk, data on overnight stays in the summer seasons between 1997 and 2018 in Austria were considered [17]. These numbers remained constant at approximately 60 million between 1997 and 2007 and then steadily increased until 2018 by approximately 20% [17]. 

## 3. Results and Discussion

The total number of studies, reports and reliable webpage sources considered relevant for describing the death risk (mortality) and the potential risk factors was n = 42 (mountain hiking, n = 7; mountain biking, n = 7; paragliding, n = 4; trekking, n = 9; mountain, rock and ice climbing, n = 12; high-altitude climbing, n = 11).

### 3.1. Mountain Hikers

Mountain hiking is defined as walking in a mountainous environment predominantly on marked trails and paths [18]. However, boulders or snowfields can also be part of a mountain hike as well as short and very easy rock-climbing passages with or without the support of wire ropes [16]. Therefore, mountain hiking is usually performed at altitudes below 3000 m (although there are some exceptions, e.g., in the Alps or Sierra Nevada) and includes easy hikes on broad trials but also activities at the border to mountain as well as rock or ice climbing (these disciplines are described in detail in the paragraph “mountain, rock and ice climbers”).

Mountain hiking is a very popular summer sport activity in mountainous regions all over the world. In the Austrian Alps for example, several million hikers of all age groups and sexes, with and without pre-existing medical conditions, are attracted by this mountain sport activity each year [19].

Routine recordings of the Alpine Police in Austria showed 82 to 134 fatalities each year during mountain hiking in the Austrian Alps in the period from 1997 to 2018 (22 years), with an average of 110 deaths per year. Although the number of overnight stays in Austria increased by approximately 10% from 2014 to 2018 [17], the average death rate (102 deaths per year) during this period was approximately 20% lower compared to the years from 1997 to 2001 and 2002 to 2006 (123 and 119 deaths per year, respectively) [13,15]. For France, Soulé et al. reported approximately 25 fatalities per year, calculated for a 4 year period, with a slight predominance of traumatic (approximately 45%) versus non-traumatic accidents (approximately 35%) and nearly 20% disappearances [20]. In Austria, similar distributions of the causes of death have been reported, with sudden cardiac deaths (44%) and the consequences of falls (46%) responsible for approximately 90% of all fatalities in mountain hiking [16,21] (Table 2). The risk was found to be significantly higher in males compared to females [16,21]. Not surprisingly, the sudden death risk is steeply increasing with age [1,21]. Appropriate individual health and fitness status are considered important preventive factors [1,16,21]. 

Unfortunately, an exact death risk calculation is not possible for mountain hikers because the population at risk and exposure times can only be roughly estimated. For example, a death risk during mountain hiking in the Austrian Alps has been estimated at approximately 0.04 deaths per 1000 persons and year [1] (Table 1) corresponding to 5.7 death per 1 million hours of exposure when assuming an annual average of seven hiking days for each person [22]. It has to be mentioned that these calculations were based on hikers with Austrian nationality, which may not be representative for all hikers (including tourists) in Austria. In addition to the number of overnight stays in Austria, the 47% increase in the number of members of the Austrian and the German Alpine Association from 2006 to 2014 could help to roughly estimate the increasing number of people at risk [16]. When relating these numbers to the development in absolute death rates, a risk reduction for fatalities of approximately 30% to 40% may be assumed for the past 5 year period. Thus, when compared to the annual death rate of 0.04 per 1000 hikers from 1986 to 1992, this number may have declined in recent years to approximately 0.02 per 1000 hikers. This favorable development may at least partly be due to targeted information campaigns, accelerated rescue operations (helicopters) and emergency medical treatment and intensified specific preventive efforts, e.g., online platforms of the Alpine Clubs aimed to inform the public about risk factors and preventive measures [12].

### 3.2. Mountain Bikers

Mountain biking (MTB) is characterized by cycling off road, often over rough terrain, using specially designed bikes [36]. It is an Olympic sport and a very common recreational activity, especially in the mountains during the summer season. In the Alps, it has recently been estimated that there are approximately 18.7 million mountain bikers (locals and visitors) [37]. The number of participants seems to rise worldwide: e.g., in Germany it increased by 9.6% from 2014 to 2018 [38] and in the US this figure was 23.3%, increasing from 6.75 million in 2006 to 8.32 million in 2015 [39].

Only a few studies reported data on mortality during mountain biking [40,41,42]. In particular, Kim et al. (2006) reviewed serious MTB injuries requiring trauma center admission during a 10 year study conducted in the three trauma centers of the Greater Vancouver area. From 1992 to 2002, one patient out of 399 mountain bikers died in the hospital [24], which corresponds to a case fatality rate of 0.25% (2.5 out of 1000 injured mountain bikers) [24]. Considering a total number of estimated 56,000 mountain bikers in the bike park and on the trails of the area served by the three hospitals [24], a mortality rate of approximately 0.0002% (2 out of 1 million mountain bikers) can be estimated. In Austria, the absolute number of deaths (3–6 per year) tended to increase from 2013 to 2017 [15], probably reflecting an increase in the participation in MTB. Considering that in the summer months of 2014 approximately 824,000 tourists and 600,000 Austrians practiced mountain biking in Austria [43], the absolute number of 25 deaths in the 5 year period 2013–2017 [15] corresponds roughly to a mortality rate of 0.00035% (3.5 deaths per 1 million mountain bikers) (Table 1). 

The most serious MTB injuries and deaths typically involve a rapid deceleration that results in the rider being vaulted forward over the handlebars, usually while riding downhill at high speeds [41,44,45,46], or happen during planned events such as jumps or tricks [24]. The few mountain biking deaths which have been reported in the literature exhibit two main injury locations and different organs being involved: cranio-cervical and trunk region (liver, diaphragm, lung, chest, and coronary vessel) [44,45].

Factors favoring severe falls with the risk of death include the type of riding surface, lack of familiarity with the terrain, and the surrounding environment [24,45,47] (Table 2). Unpublished data from the Austrian Alps (April 2006 to September 2017) showed 74 fatalities (1 woman and 73 men), with a mean age of 54.1 ± 13.5 years [13,15]. Cardiovascular events were the most frequent cause of fatality (52.7%), followed by accidents (falls, crashes, and collisions) (33.8%) [15]. Whereas younger bikers (<40 years) rather suffered from traumatic events, older individuals (≥ 40 years, especially men) might be more at risk for cardiovascular events. 

Considering the growing popularity of the MTB sport, recently paralleled by a continuous rise in the use of electric mountain bikes, effective preventive strategies seem to be of utmost importance. Such strategies may include improvements in safety equipment, offers in rider training, specific education, and preparatory fitness training [44,48]. 

### 3.3. Paraglider Pilots

Paragliders are free-flying, foot-launched and usually but not exclusively non-motorized aircrafts which were developed from parachutes [49,50]. The paragliding pilot is attached to a harness approximately 8 m below the wing in a sitting or semi-recumbent position and is able to achieve speeds surpassing 100 km/h [49,51]. The beginning of paragliding is dated around the early 1980s in Europe with a rise in popularity until the end of the 1990s [49,51]. No absolute participation numbers of paragliders exist and the only mortality rates found in paragliding were calculated based on the paragliding licenses in Germany [25,52]. Licensed German paragliders are primarily male (86%) and show an age distribution from 16 to 69 years, with most pilots (approximately 50%) being in the age of 30 to 39 years [25]. A high estimated number of unrecorded paragliding pilots and unreported injuries might limit the validity of these studies due to an insurance policy where most insurances do not cover injuries caused by paragliding [52]. 

Between 1997 and 2002, 7–15 deaths per year occurred in Germany. With an estimated number of active people of 16,296 in the year 1997 and 25,752 in 2002 (number of German paragliding licenses), relative numbers of fatalities ranged between 0.06% and 0.04% [25] (Table 1). Airborne sports fatality rates from the Austrian Alps resemble absolute mortality numbers from Germany reporting 5–18 deaths per year during the years 2003–2017 with an annual mean fatality rate of 9.7 deaths [13,15]. However, numbers of paragliding deaths are estimated to be lower since multiple airborne activities (e.g., hang gliding) are included in the Austrian data. 

The major cause of fatal accidents is the collapse of the paraglider (32.5%) followed by an incorrect handling of the situation, which led to an impact of the pilots with either an obstacle or the ground (59.9%) (Table 2). The incorrect use of the break lines resulting in a stall (sudden loss of lift) caused 30.1% of collapses [52]. Further causes of accidents have been found to be pilot errors (13.9%), collision with an obstacle (12%) and mistakes during take-off (10.3%) and landing (13.7%) [53]. Weather conditions seem to influence paragliding accidents, with 32% of the accidents happening during a strong wind and 35% in thermal conditions. Paragliding in foehn conditions, thunderstorm, an upcoming cold front, strong and gusty winds as well as rain should be avoided by all means [25]. Paragliding rescue is mostly performed by helicopters due to the remoteness of the paraglide locations. Since landing of the helicopter is at most sites impracticable, paragliders are a) rescued by the rescue team which walks or climbs up to the paraglider or b) by a long-line rescue (e.g., up to 200 m) in order to avoid downwash by the rotor blades [51]. 

Although pilots with less than 100 flights were the most accident-prone group (40%), the first two years after gaining the pilot license were considered the most dangerous ones irrespective of the number of flights [52]. Thus, flying skill training represents one of the most important prophylactic measures and should be regularly conducted in order to reduce the rate of fatal accidents. Further risk-reduction recommendations include appropriate equipment choice, prior pre-flight check including weather conditions and the use of protection systems [52].

### 3.4. Trekkers

The aim of trekking is to explore the mountain environment without the ultimate goal to summit a mountain [53]. Characteristic for trekking is that trekking is predominantly performed on dry ground rather than on glaciated terrain [53] and on rather well prepared tracks. Similar to the various other mountain sports activities, the number of trekkers is steadily increasing [9,54]. 

The mortality rate among trekkers is significantly lower compared to high-altitude climbers [9]. Mortality rates are well documented for the popular trekking routes in Nepal. Between 1984 and 1987, 148,000 persons obtained a trekking permit in Nepal and 23 fatalities were reported during this time period resulting in a calculated mortality rate of approximately 0.15 per 1000 trekkers [55]. A follow up study performed between 1987 and 1991 came to a similar death rate of 0.14 per 1,000 trekkers [26] (Table 1). Assuming an average trekking length of 14 days per trekker in Nepal, the risk is approximately 11 deaths per 1 million days of exposure [22]. 

The main causes for the fatalities were trauma (30–48%), illness (35%) and altitude sickness (13%–25%) [26,55] (Table 2). Trauma was mainly caused by falls. Illnesses include myocardial infarction, which accounted for approximately 30% of the illness-related fatalities, diabetic acidosis (20%) and infectious diseases (15%). The remaining fatalities were caused by hypothermia, cerebral vascular incidences and sudden cardiac death [26]. Dying from altitude sickness was mostly linked to lung and brain edema, but also to thrombosis and hemorrhage [56]. It is worth noting that a more recent analysis of the deaths caused from altitude illness during trekking in Nepal found a higher mortality rate of 0.08 per 1000 trekkers [57] compared to the former reports of 0.02 and 0.04 per 1000 trekkers [26,55]. The analyses further showed that the mean age increased from 35 ± 13 years between 1984 and 1987 to 44 ± 16 years between 1987 and 1991 to 50.9 ± 13.7 years between 1999 and 2006, possibly indicating that the average age of trekkers is increasing [26,55,57]. Interestingly, the altitude level seemed not to have an influence on the overall mortality rate during trekking in Nepal [26], which seems to indicate that the main risks of death during trekking are not related primarily to altitude but to the hazardous terrain and to illness occurring in a remote area [58]. 

As mentioned, mortality rates during mountain trekking in Nepal are well established but scientific scrutiny is limited for other countries. Looking at deaths occurring in national parks, somewhat lower mortality rates have been reported. The average mortality rate in the Yu-Shan National Park (with altitudes ranging from 300 to 3952 m), Taiwan, was found to be 0.0024 deaths per 1000 visitors and 0.05 deaths per 1000 trekkers or climbers [27]. Of note, in this analysis, no differentiation between trekkers (sometimes also called hikers) and climbers was made [27]. Similar overall death numbers were found for California’s national parks, with 0.0026 deaths per 1000 visits. However, when only considering deaths during trekking, the mortality rate was 0.00023 per 1000 trekkers [28]. 

Overall, considering the low mortality rates, mountain trekking seems to be a relatively safe mountain activity. Measures to further reduce mortality rates include training of mountain hiking skills to prevent falls, prevention of severe altitude illnesses and pre-screening for pre-existing diseases. 

### 3.5. Mountain, Rock and Ice Climbers 

Mountain, rock and ice climbing contain various sub-disciplines where potential hazards vary considerably [59]. Mountain climbing covers a set of climbing activities needed for ascending a mountain often in mixed terrain (i.e., rock, ice, snow) and in remote areas [59]. Approaching the route, route finding, placing protection material over multiple pitches and descending is usually more demanding and more autonomously compared to other sub-disciplines. Mountain climbing (also referred to as mountaineering) might be conducted in higher altitude (compare chapter high-altitude climbers for death rate and risks), but is not limited to altitudes above 5500 m. In traditional (alpine) rock climbing, the altitude is usually low or moderate and the exposure to ice is minimal [60]. Primarily, protection material needs to be placed autonomously (i.e., no or few pre-placed bolts). Sport climbing is conducted on rock outdoors and consists of lead (ascending the route and connecting the rope and pre-placed bolts with quickdraws for protection), top rope (belayed using pre-placed bolts on the top as a turning point of the rope), and bouldering (climbing without a rope to moderate heights and falling/jumping off the wall, usually protected by crash pads). Distinctive to alpine rock climbing, sport (and indoor) climbing usually consists of a single pitch. Indoor climbing similarly consists of lead, top rope, and bouldering, but is conducted on artificial walls [60]. Furthermore, speed as a special form of top rope climbing is included (ascending as quickly as possible and belayed top rope). Competitions in climbing contain lead, speed and bouldering, which all will be part of the Summer Olympic Games in Tokyo 2020 [61]. Ice climbing is conducted on icy formations (e.g., frozen waterfalls and mountain faces covered with ice) and with the help of crampons and ice axes [60]. Ice screws are used as protection material and connected to the rope using quickdraws. 

The risk factors and causes (and consequently mortality rates) may vary largely between the sub-disciplines. Hazards are largely dependent on the conditions and include rock and ice fall, roped falls and hitting the rock/ice, broken holds, avalanches, falling into crevasses, and sudden weather changes. Causes for fatalities include trauma, avalanche burial, cold injury, and sudden cardiac death [9] (Table 2). Fatalities seem to increase with increasing altitude [58] and further risk factors include lead climbing, higher length of falling, snow- or ice-covered terrain, and rappelling [30,62]. Fatalities during traditional (alpine) rock climbing are caused by head injuries or hypothermia [63]. In both indoor and outdoor climbing, falls off the wall (in the rope or on the floor) are usually harmless and frequent. External risk factors (e.g., rock fall, avalanches, weather changes) are almost non-existent in indoor climbing and minimal in outdoor climbing [60]. Hazards for ice climbing are similar to mountain climbing and additionally include collapsing of the ice formations.

Detailed figures of the mortality rates specifically for the different sub-disciplines and comparable in format are lacking for rock, mountain and ice climbing [60]. Furthermore, it is difficult to assess absolute numbers of individuals exposed, since not all climbers are organized in associations or register the summit attempts or overnight stays. The following figures were found in literature (Table 1). For mountain climbing (focus on altitudes below 5500 m), 0.13 per 1000 h of practice or 0.6 per 1000 climbers was reported (including traditional climbing, Grand Tetons, <4200 m, US) [30]. On Mt Rainier (4392 m, US), an estimated mortality rate of 0.3 per 1000 climbers was recorded [29]. On Mt Cook (3724 m, NZ), 1.87 deaths per 1000 exposure days were reported [64]. Eleven deaths were found during traditional climbing on Half Dome (2694 m, US) over a period of 85 years [65]. Thirteen fatalities were recorded in 3.5 years in the Yosemite National Park (<4000 m, US) and 6% of the rock climbing injuries registered at the local medical clinic or rescue team were fatal [63]. On average, one death per year is reported for the Canadian Alpine Club in ice climbing [60]. For the Austrian Alps, figures are available between 1997 and 2017, where on average 20 deaths during rock climbing (alpine climbing and outdoor sport climbing) and one death during ice climbing per year occurred. Out of all deaths in the Austrian Alps, 6.6% occurred during rock and ice climbing. The trend seems to decline with a 5 year average of 21 deaths between 1997 and 2001 and 16 deaths between 2013 and 2017. [13,15] As an approximation to the individuals exposed, in the period of 2005 to 2016, the number of members of the Austrian Alpine Club increased from 317,000 to 521,000 [66]. Burtscher and colleagues estimated a mortality rate in rock and ice climbing of 6.77 per 100,000 persons annually for the Austrian Alps [1]. No fatalities in both indoor and sport climbing were reported in previous studies. Although deaths occur during indoor and sport climbing, these are rare and some deaths might be connected to pre-existing comorbidities or misadventure [60]. Research in indoor and outdoor climbing focused predominantly on injuries [67,68].

Overall, mountain climbing, ice and traditional climbing seem to be the most relevant sub-disciplines for fatal incidents in rock and ice climbing. However, at least for the Austrian Alps, these incidents seem to decrease over the last 25 years; especially, when a potential increase in participation is considered. Recommendations to minimize the risks include helmet use, appropriate clothing, and attention in mixed terrain and during rappel. Fatalities in sport and indoor climbing are rare; nevertheless, risk-minimizing actions, e.g., partner-check of harness, belay device, rope (knot of the climber and knot at the end of the rope to avoid slipping of the rope through the belay device) is recommended. 

### 3.6. High-Altitude Climbers

Altitudes above 5500 m are defined as extreme altitudes where the inspired oxygen pressure falls below 50% of the sea level value associated with inability of permanent human living [69]. For a limited period, however, humans may reside in those altitudes for work or climbing activities without the use of supplementary oxygen, even up to the top of the world (Mount Everest, 8848 m) [70]. The number of climbers attracted by mountains in extreme altitudes is steadily growing, at least partly due to cheap flight options, the construction of new roads and paths, and commercial services available for many destinations in such extreme destinations [71]. 

What often remains neglected, however, is the fact that those regions are mostly inhospitable areas, where steep rocky, snowy and/or icy terrains with poor accessibility, and unpredictable risks of avalanches, falling rocks and ice is aggravated by severe hypoxic and cold environments. Inspired oxygen pressure at an altitude of 5500 is only half and on the top of Mount Everest (8848 m) not more than 30% of that at sea level [72]. Ambient temperature drops by approximately 6.5 °C per 1 km gain in altitude [73], and weather and track conditions can change dramatically within hours. These tremendous objective threats may be compounded by subjective hazards such as insufficient physical fitness and health status, lacking mountaineering skills and/or inappropriate equipment [22]. Not surprisingly, depending on the presence and combination of these conditions, variable high risk for fatal accidents and emergencies may result, frequently also as a consequence of the difficulties in managing medical interventions and rescue operations. 

Reported mortality rates may vary between 0.03% (3 death per 10,000 climbers; Kilimanjaro) and over 4% (40 deaths per 1000 climbers above basecamp; Annapurna I). Kilimanjaro and Aconcagua are among the most popular mountains, with elevations higher than 5500 m. Approximately 30,000–40,000 climbers attempt to reach the top of Kilimanjaro (5895 m, Tanzania, East Africa) each year [31,32]. It is difficult to ascertain the exact number of people who die on Kilimanjaro each year. An attempt to obtain accurate numbers was made between 1996 and 2003, where 25 deaths have been recorded on Kilimanjaro [33]. However, this number does not necessarily include porters and other mountain crew [33]. Assuming that there are twice as many climbers during these years as there are now and assuming that unrecorded porters and crew members are among the deceased, the number of people who die was estimated to be approximately 10 per year [32,33], which corresponds to a mortality rate of 0.03%. Most deaths are due to high-altitude illness [31]. Between 2001 and 2012, 42,731 climbers tried to summit Aconcagua (6961 m, Andes mountain range, South America) and 33 subjects died during this period (mortality rate: 0.08%) [34]. Deaths were caused by falls (24%), high-altitude illness (21%), hypothermia (15%), sudden cardiac death (12%) and other causes (28%, Table 2). Higher mortality rates (0.3%) were observed on Denali (Mount Mc Kinley, 6190 m, Alaska), where 96 died out of 31,201 climbers (93% males) who attempted to reach the summit (52% success rate) from 1903 until 2006 [53]. Falls (44%) were the predominant cause of deaths, followed by exposure/hypothermia (16%), and high-altitude illness (7%). The authors found a 53% decrease in death rates when comparing the periods from 1903 to 1994 and 1995 to 2006. The death risk increases when climbing 8000er peaks (climbers above base camp), ranging between 0.64% (Cho Oyu, 8188 m, Himalayas), 1.56% (Mount Everest, 8848 m, Himalayas) and 4.5% (Annapurna I, 8091 m, Himalayas) [35,58] (Table 1). When related only to the numbers of summiteers, these figures rise to 1.4% for Cho Oyu, 6% for Everest and 38% (!) for Annapurna I, the most dangerous 8000er because of the extremely high risk of avalanches [74]. In general, falling was the most common cause of death (41%) followed by avalanches and exposure, high-altitude illness, and other causes [35]. 

The selection of low-risk 8000 m mountains, improved logistics services, appropriate acclimatization strategies and medical advice, the use of modern equipment and optimized weather forecasting, might all contribute to reduced mortality rates, which may be the reason for somewhat reduced odds of deaths during the last decades [22]. Interestingly, it was shown that experience from previous climbs in the Himalayas had no beneficial effects [35,75]. In this context, it is important to recognize that individual fitness may change within weeks/months but the acquisition of appropriate mountaineering experience (climbing skills, danger assessment, etc.) takes years [22,76]. Climbing at extreme altitude remains a high-risk adventure, which can be modified by appropriate individual preparation, selecting lower-risk mountains and the use of modern logistic services.

### 3.7. Limitations

The following limitations have to be acknowledged. Considering the limited number of scientific work on this topic, we did not follow the procedure of a systematic review or meta-analysis (e.g., Preferred Reporting Items for Systematic Reviews and Meta-Analyses (PRISMA) Statement [77]) but included data from national databases known to the authors (not published in scientific journals) and data from webpages. Consequently, we did not account for the risk of bias within or across the sources included and followed a rather subjective approach of study selection and assessment. Additionally, absolute numbers of individuals exposed are mostly missing and thus mortality rates in this review are often based on estimated figures. Moreover, only limited information on potential risk factors for death during mountain sports activities is available. While we reported some data on the extent of death in mountain sports in the present study, future research might focus more on examining risk factors for traumatic and non-traumatic death. Lastly, death rates of different regions and environmental conditions, with different populations at risk involved, were included, which makes comparison of these rates rather difficult.

## 4. Conclusions

Death rates vary dramatically between different mountain sports activities primarily practiced during the summer season. Mortality in paragliding, mountain, rock and ice climbing, and especially in high-altitude climbing, is higher compared to that in mountain hiking, trekking or mountain biking. Risk factors as well as the characteristics of the practitioners vary across different mountain sports activities. High-risk activities are characterized by an increased likelihood of fatal falls and the presence of considerable objective hazards such as avalanches, rock and ice fall, or rapidly changing weather conditions. Appropriate skill and fitness levels are of prophylactic relevance when engaging in those activities. Lacking fitness and preexisting diseases put elderly people at special risk for non-traumatic death. The death risk is increased in remote terrains, e.g., high altitude, where rapid rescue missions are impossible. Thus, preventive efforts must more particularly focus on high-risk mountain sports and high-risk individuals.

## Figures and Tables

**Table 1 ijerph-16-03920-t001:** Selected mortality rates during different sport activities in the mountains.

Mountain Sports Activity	Region/Mountain	Mortality Risk (Death Rate per 1000 Persons at risk)	Time Period	Reference
Mountain hiking	Austrian Alps	0.04 ^a^	1986–1992	[1,23]
	Austrian Alps	0.02 ^a^	2014–2018	[15,17]
Mountain biking	Austrian Alps	0.0035 ^a^	2014	[15]
	Greater Vancouver area	0.002 ^a^	1992–2002	[24]
Paragliding	Germany	0.46 ^a^	1997–2002	[25]
Trekking	Nepal	0.14 ^b^	1984–1987	[26]
	Yu-Shan National Park, Taiwan	0.05 ^b^	1985–2007	[27]
	California’s National Parks, US	0.00023 ^b^	1993–1995	[28]
Mountain, Rock and Ice climbing	Austrian Alps	0.068 ^a^	1986–1992	[1,23]
	Mt Rainier, US	0.3 ^b^	1977–1997	[29]
	Grand Tetons, US	0.6 ^b^	1981–1986	[30]
High-altitude climbing	Kilimanjaro, Africa	0.3 ^b^	2017	[31,32,33]
	Aconcagua, South America	0.8 ^b^	2001–2012	[34]
	Cho Oyu, Himalayas	6.4 ^b^	1970–2010	[35]
	Mount Everest, Himalayas	15.6 ^b^	1970–2010	[35]
	Annapurna I, Himalayas	45.0 ^b^	1970–2010	[35]

^a^ means annual death rate per 1000 people at risk, likely doing several ventures per year; ^b^ means annual death rates per 1000 people at risk, doing the one specific venture.

**Table 2 ijerph-16-03920-t002:** Main causes and risk factors in different mountain sports activities.

Mountain Sports Activity	Main Causes	Main Risk Factors
Mountain hiking	Non-traumatic (mostly cardiac) death	Male sex
	Trauma-related death (falls)	Higher age
		Pre-existing diseases
		Insufficient physical fitness
		Inappropriate equipment
Mountain biking	Trauma-related death (falls, crashes)	Lack of familiarity with the terrain
	Non-traumatic (mostly cardiac) death	Lack of skill
		Risky behavior Pre-existing diseases
Paragliding	Trauma-related death (falls)	Collapse of the paraglider
		Incorrect use of the break lines
		Strong wind
		Lack of skill
Trekking	Trauma-related death (falls)	Higher age
	Non-traumatic (mostly cardiac) death	Pre-existing diseases
		High-altitude illness
		Insufficient physical fitness
Mountain, rock and ice climbing	Trauma-related death (falls, rock/ice fall)	Lack of skill
	Avalanche burial	Insufficient physical fitness
	Non-traumatic (mostly cardiac) death	Inappropriate equipment
High-altitude climbing	Trauma-related death (falls, rock/ice fall)	Exposure
	Hypothermia	Insufficient physical fitness
	Avalanche burial	Lack of skill
	High-altitude illness	Inappropriate equipment
		Pre-existing diseases

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
