# Peer review of "Mortality in Different Mountain Sports Activities Primarily Practiced in the Summer Season—A Narrative Review"

_ijerph, 2019, doi:10.3390/ijerph16203920_

Round 1

Reviewer 1 Report

This is a well documented and written manuscript. 

Here are my comments:

Introduction - Please specify how knowing the mortality rates would be helpful for sports participants. Material and methods - Please specify the process of selecting studies and describe the method of data extractions and handling. - How you handle the risk of bias that may affect the cumulative evidence? Results - How many studies were screened, assessed for eligibility and included in the review - for all and for each mountain sports studied in this review? Please discuss the limitations of the study (at review level, outcomes level).

Reviewer 2 Report

Thank you for the opportunity to review this manuscript. The present study aimed to summarize the differences in mortality rates and potential risk factor of mountain sports activities. It could be highly evaluating focusing point but some methodological issues exist.

Reviewer 3 Report

This is a well-organized and well-written systematic review regarding risk factors for death and injury during mountain sports activities. While this is an important review within the field, it does not necessarily have wider impacts, limiting its potential reach.

Major comments:

The review mostly reports the numbers of injuries or fatalities, whereas risk factors are harder to determine. The paper would have more value if there could be a greater emphasis on risk factors. Otherwise, this seems to merely be a report on the number of injuries and fatalities, and the import of such data is unclear. Some paragraphs throughout the manuscript are lengthy and thus difficult to follow. Each paragraph should be begin with a topic sentence, and all subsequent sentences in the paragraph should support the topic sentence. Please follow this guide in re-paragraphing the paper, as some paragraphs run almost a full single-spaced page and could easily be broken into several smaller paragraphs. It would be helpful if a table could be created that compared the mortality rate among different mountain sports activities. I realize that none of the studies reported made a direct comparison, but such a table would better help to unify the data.

Minor comments:

Line 24 contained some unclear language: "to that during paragliding, rock and ice or high-altitude climbing". I believe you mean "to that during paragliding or during rock, ice, or high-altitude climbing." Line 25: define "young people" Line 48-49: The wording suggests that the literature search was performed during this time period. Instead, I believe you mean "using a cutoff date of June 2019" rather than "until June 2019". Line 55: Rather than "Additional", I believe you mean "Additionally". Line 55-56: As "data" is a plural verb, the verb for this sentence should be "were" rather than "was". Line 71: Similarly to that described above for Line 24, the commas in this sentence make it confusing. I believe you mean "...at the border to mountain as well as rock or ice climbing..." In addition, the word "boarder" should be "border". Lines 113-114: The wording here is confusing, as it suggests the bikers have been estimated, whereas it is the number of bikers that has been estimated. I suggest that you word this thusly: "In the Alps, it has recently been estimated that there are 18.7 million mountain bikers." Moreover, please clarify whether these are individuals living in the Alps who are bikers or if this is the number of bikers who visit the Alps annually to engage in biking. The "%" symbol is used inconsistently throughout the paper. In some cases, it immediately follows the number, whereas in other cases, there is a space after the number before the % symbol. Accepted practice typically has the % symbol immediately following the number (50% rather than 50 %). Please make this correction throughout the paper.

Round 2

Reviewer 2 Report

Thank you for your conscientious revision.

After reading the resubmitted revised version of the manuscript, I think the authors are well reflecting my comments as a reviewer.

Could you explain just one point? That is about the purpose of describing the secular trend.

Introduction

Paragraph three and four. I think that you well explain the importance of describing the knowledge on mortality of each mountain sports. But you should explain the social and academic significance of describing the secular trend. How does exposing the secular trend contribute to practitioners and researchers.

Author Response

Dear Reviewer,

thank you again for your valuable suggestion.

We know added as follows (line 52-54):

… or due to preventive efforts, e.g. initiated by mountaineering associations [12], the information on the year, when the data was collected, should be considered. Describing secular trends may help to evaluate potential effects of such changes/measures.